# Quality of Life in Breast Cancer Survivors in Relation to Age, Type of Surgery and Length of Time since First Treatment

**DOI:** 10.3390/ijerph192316229

**Published:** 2022-12-04

**Authors:** Sergio Alvarez-Pardo, Ena Monserrat Romero-Pérez, Néstor Camberos-Castañeda, José Antonio de Paz, Mario Alberto Horta-Gim, Jerónimo J. González-Bernal, Juan Mielgo-Ayuso, Lucía Simón-Vicente, Jessica Fernández-Solana, Josefa González-Santos

**Affiliations:** 1Department of Sports, Isabel I University, 09003 Burgos, Spain; 2Division of Biological Sciences and Health, University of Sonora, Hermosillo 83000, Mexico; 3Institute of Biomedicine (IBIOMED), University of León, 24071 León, Spain; 4Department of Health Sciences, University of Burgos, 09001 Burgos, Spain

**Keywords:** breast cancer, mastectomy, breast-conserving therapy, quality of life, age

## Abstract

(1) Background: Quality of life assessment is a critical aspect of breast cancer patient outcomes, as diagnosis, prognosis and treatment can have a major impact on quality of life. The aim of this study was to describe the characteristics of the sample and to verify the relationship between quality of life (QOL) in women diagnosed with breast cancer (BC) and their age, type or surgery and time since treatment; (2) Methods: a cross-sectional, descriptive study was conducted with 183 women diagnosed with BC, aged 30–80 years in Mexico. Women’s QOL was assessed using two questionnaires, The European Organization for Research and Treatment of Cancer-Quality of Life Core Questionnaire (EORTC QLQ-C30) and The Breast Cancer Module (EORTC QLQ-BR23). (3) Results: the results show significant differences in several items when the variable age is taken into account, indicating that younger women have poorer social and sexual function, as well as poorer sexual enjoyment and lower expectations of the future. The type of surgery also indicates a significantly better QOL in those women who receive conservative treatment versus a mastectomy; the time elapsed since surgery does not show any significant results, except for sexual functioning and breast-related symptoms where >5 years implies better scores on the items. (4) Conclusions: in conclusion, it is important to take into account the characteristics of women with BC, particularly at the time of treatment, in order to mitigate the impact of the disease on their QOL with the help of a multidisciplinary team.

## 1. Introduction

According to the World Health Organisation (WHO), breast cancer (BC) is the most common and most frequent cancer among women and was the fifth most deadly cancer in the 2020, with an estimated of 2.26 million new cases and 685.000 deaths [1]. It is considered a public health problem in both developed and developing countries because of its frequency and the amount of health service resources needed for detection and treatment [2].

Currently, BC affects 12% of women worldwide, or 1 in 8, with a 5-year survival rate of 90% [3]. In Mexico, BC was registered as the second most common type of cancer in the population, after prostate cancer, although it represents the highest incidence in the female population, accounting for 25% of new cases. In addition, it is considered the leading cause of cancer death in women in this country, affecting mostly women between the ages of 50 and 60 [2,4].

In recent years, there has been significant progress in the early diagnosis and treatment of BC, reflecting significant improvements in survival rates through the adoption of best practices for cancer management [5]. However, due to the increased rates of BC survivorship, it is important to add a fundamental aspect to patient outcomes and day-to-day life, the assessment of quality of life (QOL). QOL is an aspect that needs to be considered in the context of any oncology safety assessment [6]. Due to major advances in early detection and adjuvant therapy in recent years, life expectancy has also been prolonged and QOL issues have become particularly relevant [7].

The most common treatments accepted as valid techniques in BC patients refer to breast-conserving treatments (BCT) and mastectomies (MST) [8]. Several studies have compared these two techniques but with discordant oncological results in some aspects such as survival, although they showed equivalent improvements in systemic treatment, local and distant recurrence [9,10,11]. BCT is a breast-conserving surgery which seeks to preserve the breast while respecting oncological principles in order to provide a greater aesthetic [6]. However, MST, considered as an invasive surgical treatment, entails physical changes that may have adverse effects on women’s self-esteem and intimate relationships, affecting patients’ body image, mood and/or QOL [12].

Therefore, for many oncological practices, a detailed assessment of QOL was previously performed [10], as several factors can directly affect QOL, from psychological to physical factors, and caused by the strong symptomatology experienced by more than 61% of patients up to 5 years after diagnosis [13,14]. On the physical side, patients have to deal with the possible long-term side effects of cancer care, such as pain, vomiting, risk of infections, insomnia and fatigue, which are among the most common symptoms [14,15], with fatigue being the one that lasts the longest and affects people the most [16], appearing in 1 out of every 4 diagnoses [17]. On the psychological side, the diagnosis, change in lifestyle or social relationships may contribute to an increase in negative emotions such as depression and anxiety [15]. More commonly, younger patients have a higher risk of anxiety, depression and intrusive thoughts, and 20–30% of them experience fear of the future related to the perceived risk of disease recurrence and death [18,19]. It is often the discrepancy between the patients’ current image and the ideal image created by society that often leads to dissatisfaction and emotional distress, fostering a preoccupation with external appearance [15,20]. It should also be noted that fear or worry about cancer recurrence (FCR) is a factor with great psychological influence, showing that higher levels are related to worse QOL [21]. Another factor of great importance is the social support received by BC patients. Social support is a fundamental human need crucial for adaptation to the disease, which decreases distress and depression, and reduces the level of recurrence, thereby positively influencing patients’ QOL [22].

Another complex and important area is sexuality, which includes psychosocial, socio-cultural and biological aspects. Sexual dysfunction can be frequently observed in patients with BC, in addition to the negative impact that treatment can have. Therefore, satisfaction with sexual life also becomes a critical issue to include in the QOL of patients with BC and should be included in the individual assessment [18]. Of note is the identification of patients under 50 years of age as a vulnerable group as they have worse survival, more severe psychosocial effects and due to the convergence of menopause with BC, worse sexual functioning [23,24].

However, it should be noted that most of the studies that have been conducted on QOL in BC have been carried out in Western countries, with relatively few from other parts of the world. In Latin America, the age-standardised incidence rate is still better than in Western Europe, although survival rates are lower and 30–40% of diagnoses are metastatic due to late diagnosis and poorer access to treatment [25].

It is important to highlight that higher QOL is associated with higher long-term survival [26,27], where the 5-year survival rate for non-metastatic BC is 90% and the 10-year survival rate is 84% [28]. The aim of this study was to describe the characteristics of the sample and to verify the relationship between QOL in female BC survivors and their age, type of surgery and time since first treatment.

## 2. Materials and Methods

### 2.1. Participants

This cross-sectional, descriptive study was conducted in the state of Sonora, Mexico. The sample consisted of 183 women with a diagnosis of BC aged between 30 and 80 years. They were divided into two groups with a cut-off age of 50 years as this is the median age of menopause in Mexico [29,30] and the rest of the world [31]. Likewise, this age cut-off is widely used in the literature where variables associated with QOL are studied [32,33,34,35].

Convenience sampling was performed since the women chosen for the study frequently attended consultations at different hospitals in the state. All of them participated voluntarily and provided written informed consent after explanation of the nature and intention of the study; they fulfilled the inclusion criteria: (1) have a cancer diagnosis, (2) have had surgery as part of the treatment (radical or conservative), (3) have a clear understanding of the language, (4) possess the ability to read and write for answering the questionnaire and (5) were 18 or older when answering questionnaires.

The main exclusion criteria was not to be undergoing chemotherapy (CTX) or radiotherapy (RT), not being pregnant during the QOL assessment and having undergone a MST following breast reconstruction.

### 2.2. Procedure

The study was approved by the Bioethics Committee on Human Research of the Department of Medicine and Health Sciences of the University of Sonora (DMCS/CBIDMCS/D-50). All women who participated in the study came from different oncology speciality hospitals in the province of Sonora. The evaluation of subjects was carried out in two stages. For the first evaluation, data were taken on two occasions, previously carrying out a questionnaire of our own elaboration which was validated by experts in the field. Subsequently, at the second stage of the evaluation, QOL was measured using two questionnaires: The European Organization for Research and Treatment of Cancer–Quality of Life Core Questionnaire (EORTC QLQ-C30) and The Breast Cancer Module (EORTC QLQ-BR23): both questionnaires have been obtained from the Quality of Life Unit of the Data Centre of the European Organisation for Cancer Research and Treatment EORTC in Brussels, Belgium. None of the questionnaires completed by the participant population were excluded as all individuals in our sample answered more than 70% of the total number of items [36].

### 2.3. Assessments

The most commonly used questionnaire for the measurement of QOL in BC patients, according to the literature, was the EORTC QLQ, which is used alone or with specific modules such as BR23, although they are most often used together [37,38].

The EORTC QLQ-C30 contains a scale which includes symptoms such as pain, fatigue, nausea and vomiting, dyspnea, sleep disturbance, constipation, diarrhea, appetite loss and financial difficulties scores (13 items: nausea and vomiting, pain, dyspnea, sleep disturbance, appetite loss, constipation, diarrhea and fatigue), and a functional scale with social functioning, role functioning, physical functioning, cognitive functioning and emotional functioning scores (15 items: strenuous activities, self-care, long/short walk, limitations at work, limitations in leisure, depression, worry, tension and irritability). Each item is rated on a scale from 0 (not at all) to 4 (very much), with the exception of two items on the global health/QOL scale which use modified 7-point linear analog scales [39]. The EORTC QLQ-C30 has been validated for a large number of European countries and the rest of the world [40] and especially for the Mexican population [41].

The EORTC QLQ-BR23 consists of 23 items, and is a tumour-specific tool, which incorporates four functional scales—body image, sexual functioning, sexual enjoyment and future perspective—and four symptom-oriented scales—systemic therapy side effects, breast symptoms, arm symptoms and upset by hair loss scores. These items belong to side effects related to different treatment modalities, such as surgery, CHTH or RTH (15 items), dimensions of body image (4 items), sexuality (3 items) and future perspective (1 item). Each item is graded on a scale from 0 (not at all) to 4 (very much) [39]. All scores from 1 to 4 or from 1 to 7 have been converted and range from 0 to 100 [41]. The EORTC QLQ-BR23 has been validated for the Mexican population, as well as for other populations [41].

In both tests, a high score on a functional scale represents a high and healthy level of functioning and a high score on the global health status represents a high QOL but conversely, a high score on the scale or items of symptoms, represents a high level of symptomatology or problems resulting in a worse QOL [41].

### 2.4. Statistical Analysis

Data were shown as number of cases and (%), or as mean ± standard deviation of the mean (SD). Statistical analysis was performed with SPSS software version 25 (IBM-Inc., Chicago, IL, USA). Statistical significance was determined at a *p*-value < 0.05.

Differences in each of the items of each questionnaire between the groups formed were tested by means of a univariate test with the group to which each patient belonged as the fixed factor.

A bivariate analysis was performed using an independent Student *t*-test to examine the relationship between case incidence and groupings according to data collected at the time of assessment for age (<50 vs. ≥50), type of surgery (BCT vs. MST) and years since surgery (<5 years vs. ≥5 years).

Finally, statistical power and effect sizes were calculated. Effect sizes were determined using Ferguson’s criteria (partial eta squared (η^2^)), where there is no effect if 0 2 *p* < 0.05; minimal effect if 0.05 2 *p* < 0.26; moderate effect if 0.26 2 *p* < 0.64; and strong effect if 2 *p* 0.64 [42].

## 3. Results

The clinical characteristics of the 183 women BC survivors who participated in the study are shown in Table 1. The mean age of the women was 52.9 ± 8.6 years, and 43.2% of them were <50 years. Of the total participants, 38.8% (*n* = 71) of them underwent BCT, 56.6% (*n* = 112) MST and 7.5% (*n* = 15) underwent MST followed by breast reconstruction, i.e., 64.1% of the sample was subjected to an MST. The vast majority are married, unemployed and with a level of education up to high school. It is worth noting that a large majority (66.1%) have some family history. Furthermore, 75.4% of the sample had been treated less than 5 years before.

Analysis of QOL based on the EORTC QLQ-C30 between the <50 years vs. ≥50 years groups (Table 2) shows differences in items related to symptomatology, where the <50 years group have higher scores on all items except the diarrhoea item, although without significant results. The social functioning item is significant (*p* = 0.027), with younger women expressing the lowest level of social function.

In the analysis of QOL based on the EORTC QLQ-BR23 between groups <50 years vs. ≥50 years (Table 3), significant differences are shown in the sexual functioning item (*p* = 0.002), sexual enjoyment item (*p* <0.001), where the older age group reflects greater sexual function and enjoyment; in the future outlook item (*p* = 0.035), in which the group ≥ 50 years old shows less concern about their future; in the section breast-related symptoms (0.044), where those under 50 years of age report less symptomatology; and finally in the item concerning hair loss (*p* = 0.008), where more concern is reflected in the older age group.

Analysis of QOL based on the EORTC QLQ-C30 between the BCT vs. MST groups (Table 4) shows significant differences. It was observed that on the items of role functioning (*p* = 0.031) and general health (*p* = 0.001), women in the MST group had a statistically higher value than those in the BCT group. However, it was observed that women in the MST group had a statistically lower score than those who underwent BCT on the nausea and vomiting item (*p* = 0.001).

Symptoms with the highest scores between the two groups were insomnia, fatigue and pain, which were always higher in the BCT group than in the MST group.

The analysis of the QOL based on the EORTC QLQ BR23 between the BCT vs. MST groups (Table 5) shows no significant differences in the items on the scale. In this case, it can be seen that the scores on body image are statistically higher in the BCT group with respect to MST as well as the scores on the item sexual enjoyment, although without significant differences; and the BCT group was higher than the EORTC reference values 82.98 ± 22.87 in women with European BC [43].

In the analysis of QOL based on the EORTC QLQ-C30 among groups < 5 years from diagnosis to QOL measurement, vs. >5 years from diagnosis to measurement of QOL (Table 6), no significant results were found, but it can be seen that the overall health item score is higher in women who were measured >5 years after diagnosis of the disease.

On the other hand, certain items on the symptom scale improve over time, although without being significant, such as fatigue, pain, nausea and vomiting, anorexia and diarrhoea. However, there are symptoms that increase over time, such as dyspnoea, insomnia and constipation.

In the analysis of QOL based on the EORTC QLQ-BR23 among groups (the <5 years from diagnosis to QOL measurement vs. >5 years from diagnosis to QOL measurement) (Table 7), statistically significant differences were found in the sexual functioning items (*p* = 0.021) where the >5 years group reported better sexual function, and in the breast-related symptoms item (*p* = 0.005), where the >5 years group reported greater symptomatology.

## 4. Discussion

The aim of this study was to describe the characteristics of the sample and to verify whether age, type of surgery and/or time since first treatment were factors that could be related to QOL in women diagnosed with BC. Studying QOL in mastectomized women may help to improve QOL, as some cancer survivors are more likely to have worse QOL than others [44].

The way to understand this fact is to identify which variables may affect the woman negatively or whether high symptomatology is directly related to worse QOL [14].

There is evidence that approximately 92% of MST patients have three or more symptoms in the first year after diagnosis [14]. The most common symptoms are insomnia, pain, dyspnoea or fatigue. In total, 61% of survivors report these four symptoms for 5 years after diagnosis, with fatigue being the most commonly reported symptom [16], and may even be present up to 10 years after diagnosis. It can also affect functionality and mental health, increase emotional problems, aggravate insomnia and increase pain and even dysfunctional behaviours [17], which is in agreement with the present study that shows that the most frequent symptoms are insomnia, fatigue and pain.

There are some study variables that have not shown conclusive evidence of whether there is a relationship between them and QOL in BC survivors. These factors associated with QOL are age at diagnosis, where most of the scientific literature has shown that older age implies higher QOL. This may be because older survivors have less concern about physical appearance, less stress, fewer financial problems, fewer side effects, less concern about possible infertility and receive less CTX doses during treatment compared to younger survivors [23,32,45,46,47]. However, there are studies that claim that women under 50 years of age have better QOL as they have a greater social role and general support [34].

The results show that the variable age has a significant effect on the symptomatology of survivors in both the EORTC QLQ-C30 and the EORTC QLQ-BR23, where younger women have the highest scores on the scales, except for the items diarrhoea, BRBS andBRHL, with SF, BRSEF, BRSEE, BRFU, BRBS and BRHL being significant items and agreeing with other studies where it was observed that younger women receive more treatment, thus increasing the side effects [23,48]. On the other hand, the symptomatology is higher in the elderly women in BRBRS, BRAS and BRHL, agreeing with two studies where they believe that the main culture/religión and a worse libido generates more symptomatoloy in these items [34,49]. Young women show a significant correlation with the BRFU item (*p* = 0.035), where they see worse future prospects than older women, because they are usually the breadwinners or the ones responsible for the children, reporting greater concern for their future health [45,48,50]. Similarly, there are problems such as the development of lymphoedema, which can be an impediment to returning to work, making the time off work longer and therefore increasing uncertainty and mental stress [45,51]. Otherwise, older women report greater sexual function and enjoyment, with values well above other studies in the Mexican population, similar to research reporting that young BC survivors who have received some form of treatment have less sexual enjoyment due to early menopause resulting from treatment [52]. On the other hand, they reflect greater symptomatology in the BRBS item (*p* = 0.044), which it is not in line with other studies where no significance has been determined between the two variables [26,48], although it is corroborated in the study by Imran et al. [34]. In addition, younger women have different psychological and social problems that tend to contribute negatively to the QOL such as depression, weight gain during treatment, concern about early menopause and infertility, which because of treatment decreases the chances of pregnancy by 40% [32,34].

On the other hand, younger women who are diagnosed with BC have been shown to correlate with poorer body image than older survivors, which is closely related to the onset of early menopause, possible infertility and greater concern about appearance and external judgements. These changes appear to be more evident in women under 50 years of age linked to greater dissatisfaction and negative self-esteem predicting lower levels of QOL and higher levels of anxiety and depression, greater preoccupation with sexual and intimate appearance and poorer survival [53]. One aspect associated with great relevance in QOL is the type of surgery, where it is stated that women who received conservative surgery have a better QOL than those who underwent radical MST [26,54,55,56,57], although there are studies that show no significant differences [50,58,59,60]. When analysing the type of surgery, the results found in the scientific literature are similar to those of this study, where the survivors of the MST group have better QOL than the BCT, whereas others report no differences between the groups [61,62,63,64,65,66], neither in the items related to emotional aspect, anxiety, depression, mood and well-being [50].

Another related variable is the time elapsed from surgery to QOL measurement, where the more time elapsed between the two moments, the better the results, having less symptomatology [35,67] and better cognitive functioning [68]. However, the relationship between time between surgery and QOL measurement in breast survivors in our study measured with the EORTC QLQ-C30 shows no significant results, but it can be seen that those women who had their QOL measured >5 years after diagnosis have higher scores on the general health item, higher functionality and lower scores on the symptomatology scale, except for the items dyspnoea, insomnia and constipation. This is in accordance with longitudinal reports that explain how QOL can improve or even deteriorate, but not significantly, nor as a result of BC, but also due to the effect of age. Therefore, an increase/decrease in QOL or an increase/decrease in symptomatology cannot be directly attributed to the time from diagnosis to QOL measurement but must consider the effects of ageing [28]. Regarding the EORTC QLQ-BR23 scale, significant differences were found in the items sexual functioning (*p* = 0.021) and breast-related symptoms (*p* = 0.005), both reporting higher scores in the >5 years group, in agreement with both studies [26,28].

Finally, regarding the limitations of the study, it should be noted that the results cannot be generalised to the entire population of Mexico or to the world population, as the sample is taken from only one state in Mexico. It would be important to take into account the effect of culture when trying to extend the results to another population that differs from the Latino population. The use of self-reporting questionnaires may also be a limitation in the research, as these questionnaires should be interpreted with caution, even though they are questionnaires with good psychometric properties and were validated for this population. The lack of randomisation and the lack of a response rate should also be considered. Another limitation found in the study refers to the analysis of the side effects produced by aromatase inhibitors. Despite being aware of their importance, the time elapsed differently for each participant, and it was not possible to analyse each participant’s situation separately. The need for further research on this issue, which affects so many women worldwide, is raised as an important point of the research.

New work should be initiated to develop profiles of women who are more likely to have a worse QOL after diagnosis, during treatment or after its completion, identifying the factors associated with the quality of life of these women and thus establish a personalised and individualised action protocol so that those women who have negative associated factors can receive appropriate care as soon as possible to improve their QOL and therefore increase the chances of cancer survival. These protocols may include the prescription of personalised physical exercise, therapies with professionals in psycho-oncology or workshops to improve self-care.

## 5. Conclusions

This study concludes that age, type of surgery and years from treatment to QOL measurement are variables related to various aspects of QOL in breast cancer survivors that should be taken into consideration by the health care system and included in a multidisciplinary treatment plan and comprehensive approach.

The variables studied can provide great added value to decision-making by the health system to which the patient belongs and by the patient herself, with decisions that interfere as little as possible with her QOL and creating a support network around her, both physical and psychological, depending on the patient’s characteristics.

More studies are needed to be carried out in any part of the world so that the treatment applied to a patient is much broader and not just limited to surgical and pharmacological interventions, where aspects such as physical activity and psychological support are taken into account and could fill the lack of information the patient has about her pathology.

Thanks to this study, the resources available to the health system to improve the QOL of those diagnosed with BC can be targeted more efficiently and effectively.

## Figures and Tables

**Table 1 ijerph-19-16229-t001:** Clinical and demographic characteristics of the study group.

Clinical and Demographic Characteristics		Total (*n* = 183)	BCT	MST
Age		52.9 ± 8.6	53.4 ± 8.0	52.6 ± 8.9
Age	<50 years	79 (43.2%)	31 (39.2%)	48 (60.8%)
	≥50 years	104 (56.8%)	40 (38.5%)	64 (61.5%)
Elapsed years	<5 years	138 (75.4%)	55 (39.9%)	83 (60.1%)
	≥5 years	45 (24.6%)	16 (35.6%)	29 (64.4%)
Family history	With family history	135 (73.8%)	51 (37.8%)	84 (62.2%)
	Without family history	48 (26.2%)	20 (41.7%)	28 (58.3%)
Marital status	With partner	121 (66.1%)	49 (40.5%)	72 (59.5%)
	Without partner	62 (33.9%)	22 (35.5%)	40 (64.5%)
Employment situation	Employee	52 (28.4%)	16 (30.8%)	36 (69.2%)
	Unemployed	131 (71.6%)	55 (42.0%)	76 (58.0%)
Level of education	Up to high school	110 (60.1%)	47 (42.7%)	63 (57.3%)
	University or higher	73 (39.9%)	24 (32.9%)	49 (67.1)
Type of surgery	BCT	71 (38.8%)		
	MST	112 (56.6%)		

Results are expressed as n (%); n—number, %—percentage; BCT: Breast-conserving treatment; MST: Mastectomy.

**Table 2 ijerph-19-16229-t002:** Mean score of all items in EORTC QLQ-C30 of the study group (<50 years vs. ≥50 years).

Scales EORTC QLQ-C30	<50 Years (*n* = 79) M ± SD	≥50 Years (*n* = 104) M ± SD	*p*	η^2^	Observed Power
General health (QL) ^a^	54.32 ± 28.58	52.72 ± 30.82	0.720	0.001	0.065
Physical functioning (PF) ^a^	87.68 ± 14.21	86.28± 13.84	0.504	0.002	0.102
Role functioning (RF) ^a^	92.83 ± 12.71	95.35 ± 14.21	0.215	0.008	0.236
Emotional functioning (EF) ^a^	80.59 ± 20.27	85.10 ± 17.09	0.105	0.014	0.367
Cognitive functioning (CF) ^a^	80. 52 ± 19.24	85.20 ± 15.88	0.073	0.018	0.434
Social functioning (SF) ^a^	82.07 ± 23.23	88.94 ± 18.52	0.027	0.027	0.601
Fatigue (FA) ^b^	19.41 ± 18.01	17, 63 ± 16.65	0.490	0.003	0.106
Nausea and vomiting (NV) ^b^	7.17 ± 18.04	3.37 ± 11.70	0.086	0.016	0.405
Pain (PA) ^b^	19.62 ± 21.14	16.03 ± 18.13	0.218	0.008	0.233
Dyspnea (DY) ^b^	9.70 ± 16.15	9.62 ± 18.39	0.973	0.001	0.050
Insomnia (SL) ^b^	24.47 ± 29.58	21.79 ± 26.18	0.518	0.002	0.099
Appetite loss (AP) ^b^	8.02 ± 20.82	5,13 ± 13.75	0.261	0.007	0.202
Constipation (CO) ^b^	16.46 ± 27.15	11.22 ± 20.57	0.139	0.012	0.315
Diarrhea (DI) ^b^	4.22 ± 14.49	8.01 ± 18.28	0.131	0.013	0.326
Financial difficulties (FI) ^b^	25.32 ± 35.89	23.08 ± 31.19	0.653	0.001	0.073

M—mean; SD—standard deviation; *p* < 0.05 significance level. ^a^ The higher the score on functional scales, the better the functionality. ^b^ The higher the score on symptom scales, the greater the symptomatology.

**Table 3 ijerph-19-16229-t003:** Mean score of all items in EORTC QLQ-BR23 of the study group (<50 years vs. ≥50 years).

Scales EORTC QLQ-BR23	<50 Years (*n* = 79) M ± SD	≥50 Years (*n* = 104) M ± SD	*p*	η^2^	Observed Power
Body image (BRBI) ^a^	80.06 ± 22.50	84.70 ± 18.47	0.128	0.013	0.330
Sexual functioning (BRSEF) ^a^	62.87 ± 28.49	74.68 ± 23.34	0.002	0.050	0.865
Sexual enjoyment (BRSEE) ^a^	51.05 ± 32.40	70.51 ± 32.28	<0.001	0.082	0.980
Future outlook (BRFU) ^a^	59.92 ± 32. 63	69.55 ± 28.67	0.035	0.024	0.559
Systemic treatment adverse effects (BRST) ^b^	18,51 ± 13.98	15,66 ± 14.27	0.179	0.010	0.268
Breast-related symptoms (BRBS) ^b^	83.97 ± 15.38	88,62 ± 15.32	0.044	0.022	0.525
Arm-related symptoms (BRAS) ^b^	80.03 ± 17.65	83.76 ± 18.31	0.167	0.011	0.281
Concern about hair loss (BRHL) ^b^	75.95 ± 32.44	86.54 ± 21.04	0.008	0.038	0.757

M—mean; SD—standard deviation; *p* < 0.05 significance level. ^a^ The higher the score on functional scales, the better the functionality. ^b^ The higher the score on symptom scales, the greater the symptomatology.

**Table 4 ijerph-19-16229-t004:** Mean score of all items in EORTC QLQ-C30 of the study group (BCT vs. MST).

Scales EORTC QLQ-C30	BCT (*n* = 71) M ± SD	MST (*n* = 112) M ± SD	*p*	η^2^	Observed Power
General health (QL) ^a^	44.60 ± 26.32	59.00 ± 30.64	0.001	0.056	0.902
Physical functioning (PF) ^a^	86.29 ± 15.65	87.26 ± 12.86	0.648	0.001	0.074
Role functioning (RF) ^a^	91.55 ± 18.23	95.98 ± 9.29	0.031	0.025	0.578
Emotional functioning (EF) ^a^	82.63 ± 20.64	83.48 ± 17.29	0.763	0.001	0.060
Cognitive functioning (CF) ^a^	82.47 ± 19.67	83.63 ± 16.08	0.664	0.001	0.072
Social functioning (SF) ^a^	84.27 ± 25.34	87.05 ± 17.57	0.382	0.004	0.141
Fatigue (FA) ^b^	20.19 ± 19.55	17.26 ± 15.56	0.264	0.007	0.200
Nausea and vomiting (NV) ^b^	9.39 ± 21.03	2.23 ± 7.91	0.001	0.055	0.900
Pain (PA) ^b^	19.72 ± 22.24	16.22 ± 17.54	0.238	0.008	0.218
Dyspnea (DY) ^b^	11.74 ± 19.60	8.33 ± 15.82	0.198	0.009	0.250
Insomnia (SL) ^b^	23.00 ± 29.05	22.92 ± 26.86	0.983	0.000	0.050
Appetite loss (AP) ^b^	7.04 ± 17.74	5.95 ± 16.87	0.677	0.001	0.070
Constipation (CO) ^b^	15.49 ± 26.92	12.20 ± 21.46	0.362	0.005	0.149
Diarrhea (DI) ^b^	8.92 ± 22.51	4.76 ± 11.72	0.103	0.015	0.371
Financial difficulties (FI) ^b^	23.94 ± 35.72	24.11 ± 31.70	0.974	0.000	0.050

M—mean; SD—standard deviation; *p* < 0.05 significance level. BCT—breast conserving treatment, MST—mastectomy. ^a^ The higher the score on functional scales, the better the functionality. ^b^ The higher the score on symptom scales, the greater the symptomatology.

**Table 5 ijerph-19-16229-t005:** Mean score of all items in EORTC QLQ-BR23 of the study group (BCT vs. MST).

Scales EORTC QLQ-BR23	BCT (*n* = 71) M ± SD	MST (*n* = 112) M ± SD	*p*	η^2^	Observed Power
Body image (BRBI) ^a^	84.86 ± 20.33	81.32 ± 20.39	0.254	0.007	0.207
Sexual functioning (BRSEF) ^a^	68.31 ± 23.43	70.39 ± 28.01	0.604	0.001	0.081
Sexual enjoyment (BRSEE) ^a^	66.20 ± 31.62	59.52 ± 34.78	0.192	0.009	0.256
Future outlook (BRFU) ^a^	64.32 ± 26.02	66.07 ± 33.48	0.708	0.001	0.066
Systemic treatment adverse effects (BRST) ^b^	18.91 ± 15.90	15.60 ± 12.88	0.124	0.013	0.336
Breast-related symptoms (BRBS) ^b^	84.39 ± 17.25	88.02 ± 14.14	0.122	0.013	0.339
Arm-related symptoms (BRAS) ^b^	82.63 ± 18.18	81.85 ± 18.08	0.776	0.000	0.059
Concern about hair loss (BRHL) ^b^	83.10 ± 26.35	81.25 ± 27.50	0.653	0.001	0.073

M—mean; SD—standard deviation; *p* < 0.05 significance level. BCT—breast conserving treatment, MST—mastectomy. ^a^ The higher the score on functional scales, the better the functionality. ^b^ The higher the score on symptom scales, the greater the symptomatology.

**Table 6 ijerph-19-16229-t006:** Mean score of all items in EORTC QLQ-C30 of the study group (<5 years from diagnosis to QOL measurement vs. >5 years from diagnosis to QOL measurement).

Scales EORTC QLQ-C30	<5 years (*n* = 138) M ± SD	>5 years (*n* = 45) M ± SD	*p*	η^2^	Observed Power
General health (QL) ^a^	51.87 ± 28.97	58.15 ± 32.10	0.221	0.008	0.231
Physical functioning (PF) ^a^	86.62 ± 14.26	87.70 13.18	0.652	0.001	0.073
Role functioning (RF) ^a^	94.57 ± 11.42	94.333 ± 18.94	0.599	0.002	0.082
Emotional functioning (EF) ^a^	82.61 ± 18.47	84.81± 19.16	0.491	0.003	0.105
Cognitive functioning (CF) ^a^	82.41 ± 17.34	85.56 ± 18.02	0.296	0.006	0.181
Social functioning (SF) ^a^	84.75± 20.58	89.63 ± 21.70	0.178	0.010	0.270
Fatigue (FA) ^b^	18.76 ± 16.86	17.28 ± 18.43	0.619	0.001	0.079
Nausea and vomiting (NV) ^b^	5.07 ± 14.61	4.81 ± 15.74	0920	0.001	0.051
Pain (PA) ^b^	18.48 ± 19.49	14.81 ± 19.53	0.275	0.007	0.193
Dyspnea (DY) ^b^	9.42 ± 16.10	10.37 ± 21.11	0.752	0.001	0.061
Insomnia (SL) ^b^	21.26 ± 27.00	28.15 ± 29.26	0.147	0.012	0.305
Appetite loss (AP) ^b^	7.00 ± 18.21	4.44 ± 13.48	0.387	0.004	0.139
Constipation (CO) ^b^	13.04 ± 23.29	14.81 ± 25.18	0.189	0.001	0.072
Diarrhea (DI) ^b^	6.76 ± 16.68	5.19 ± 17.34	0.586	0.002	0.084
Financial difficulties (FI) ^b^	24.40 ± 34.54	22.96 ± 29.15	0.802	0.001	0.057

M—mean; SD—standard deviation; *p* < 0.05 significance level. ^a^ The higher the score on functional scales, the better the functionality. ^b^ The higher the score on symptom scales, the greater the symptomatology.

**Table 7 ijerph-19-16229-t007:** Mean score of all items in EORTC QLQ-BR23 of the study group (<5 years from diagnosis to QOL measurement vs. >5 years from diagnosis to QOL measurement).

Scales EORTC QLQ-BR23	<5 years (*n* = 138) M ± SD	>5 years (*n* = 45) M ± SD	*p*	η^2^	Observed power
Body image (BRBI) ^a^	82.43 ± 19.88	83.52 ± 22.08	0.756	0.001	0.061
Sexual functioning (BRSEF) ^a^	67.03 ± 26.60	77.41 ± 23.88	0.021	0.029	0.639
Sexual enjoyment (BRSEE) ^a^	59.42 ± 32.41	70.37 ± 36.39	0.058	0.020	0.476
Future outlook (BRFU) ^a^	63.77 ± 30.53	70.37 ± 31.16	0.212	0.009	0.239
Systemic treatment adverse effects (BRST) ^b^	17.32 ± 13.75	15.56 ± 15.51	0.469	0.003	0.111
Breast-related symptoms (BRBS) ^b^	84.78 ± 15.85	92.22 ± 12.98	0.005	0.043	0.811
Arm-related symptoms (BRAS) ^b^	81.00 ± 17.05	85.68 ± 20.74	0.132	0.013	0.325
Concern about hair loss (BRHL) ^b^	83.09 ± 25.87	78.52 ± 30.28	0.325	0.005	0.165

M—mean; SD—standard deviation; *p* < 0.05 significance level. ^a^ The higher the score on functional scales, the better the functionality. ^b^ The higher the score on symptom scales, the greater the symptomatology.

## Data Availability

Data is contained within the article.

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
