# Peer review of "Quality of Life in Breast Cancer Survivors in Relation to Age, Type of Surgery and Length of Time since First Treatment"

_ijerph, 2022, doi:10.3390/ijerph192316229_

Round 1

Reviewer 1 Report

Thank you for the opportunity to review this paper. The topic is interesting and relevant, especially as many studies on the topic are from western countries with other health care conditions.

My main concern is that the authors compared 3 dichotomous groupings with respect to quality of life, without considering any covariates. The results cannot be interpreted on the basis of the applied methods. I would strongly recommend to analyze the data with different methods to give a more elaborate and detailed picture of the QoL of BC survivors in Mexico and to understand better what the needs of certain subgroups of BC survivors might be.

Here are some further comments in detail:

Abstract:

1.       The abstract should be divided into the usual subheadings introduction/background, methods, results, and discussion/conclusion.

2.       The background is very general and not linked to the study goals and methods.

3.       The results sound as if many age-related differences had been found but in fact only 2/15 scales of the C30 and 4/8 scales of the BR23 were significantly different between the age groups.

4.       The conclusion is very general/ exaggerated, as the studied variables that were found to be linked with QoL are either not changeable (age) or a change will result in other negative consequences (mastectomy).

Introduction:

5.       The introduction is very general. Information on screening, early detection, mortality etc. is not directly linked to the study question and might be summarized. The main info is that QoL is getting more relevant as survival is longer nowadays.

6.       The decision for mastectomy compared to breast-conserving therapy might also depend on age and stage so that I think survival cannot be compared directly between the two methods.

Methods:

7.       Sample: Were women with mastectomy followed by breast reconstruction excluded or would these be included in either surgery group?

8.       Assessments: The C30 scale for role functioning is missing in the description.

Results:

9.       A sample description (e.g. mean age at survey, min/max age, exact time since diagnosis, stage at diagnosis, metastasis, recurrence, education, job situation, marriage status, living situation (alone/ with children…)) is missing. Were such data assessed?

10.   In l. 155 “183 mastectomized” relates to the whole sample of which however only 112 had mastectomy if I get it correctly.

11.   In the tables, the abbreviations should be explained. The asterisks for the explanation of symptom and functioning scales might be misleading as asterisks are usually used to show statistically significant differences. Maybe superscript letters might be used instead.

12.   A simultaneous consideration of the variables in regression analyses, and maybe a more differentiated grouping of age, might lead to much more interesting and interpretable results. The age group > 50 years summarizes a wild mixture of people striving for return to work, retired people (who usually do better) and very old people (who usually do worse again due to their general condition).

Discussion/ Conclusion:

13.   As further limitations, the missing randomization/ lack of a response rate, and lack of control of covariates (as probably not assessed), should be considered.

Author Response

Mrs. Jessica Fernández Solana

Department of health sciences

University of Burgos, Paseo Comendadores s/n.

Burgos, 09001, Spain

Tel. (+34) 947499108

Email: jfsolana@ubu.es

10-11-2022

IJERPH.  Subject: Submissions Needing Revision

Dear editor.

Thank you very much for inviting us to submit our response to reviewers for our manuscript (ijerph-2014157) entitled: “Quality of life in breast cancer survivors in relation to age, type of surgery and length of time since first treatment”

We have checked our manuscript according to the Academic Editor, the reviewers’ comments and the Journal requirements. We have also responded to some comments from reviewers point by point).

We would be very grateful if you could consider our manuscript to be published in your journal.

Yours sincerely,

Jessica Fernández Solana, OT, PT

  1. Response to Reviewer 1:

First of all, we would like to express our sincere gratitude for all comments and suggestions received from the Reviewer 1. This information has certainly enriched the text for its best understanding, thank you very much indeed. We have clarified the reviewer1’s questions. We have introduced the required changes both in our answers to the specific comments and in the final manuscript V2.

Thank you for the opportunity to review this paper. The topic is interesting and relevant, especially as many studies on the topic are from western countries with other health care conditions.

My main concern is that the authors compared 3 dichotomous groupings with respect to quality of life, without considering any covariates. The results cannot be interpreted on the basis of the applied methods. I would strongly recommend to analyze the data with different methods to give a more elaborate and detailed picture of the QoL of BC survivors in Mexico and to understand better what the needs of certain subgroups of BC survivors might be.

Here are some further comments in detail:

Abstract:

  1. The abstract should be divided into the usual subheadings introduction/background, methods, results, and discussion/conclusion.
  2. The background is very general and not linked to the study goals and methods.
  3. The results sound as if many age-related differences had been found but in fact only 2/15 scales of the C30 and 4/8 scales of the BR23 were significantly different between the age groups.
  4. The conclusion is very general/ exaggerated, as the studied variables that were found to be linked with QoL are either not changeable (age) or a change will result in other negative consequences (mastectomy).

Response: Thank you for your comments. We have modified this section.

Introduction:

  1. The introduction is very general. Information on screening, early detection, mortality etc. is not directly linked to the study question and might be summarized. The main info is that QoL is getting more relevant as survival is longer nowadays.

Response: Thank you for your comments, we have added this information in the manuscript (See lines 46-50)

  1. The decision for mastectomy compared to breast-conserving therapy might also depend on age and stage so that I think survival cannot be compared directly between the two methods.

Response: Thank you for your comments. We have amended this issue (See lines 53-57)

Methods:

  1. Sample: Were women with mastectomy followed by breast reconstruction excluded or would these be included in either surgery group?

Response: Thank you for your comments. (See lines 113-114)

  1. Assessments: The C30 scale for role functioning is missing in the description.

Response: Thank you for your comments, we have added this information (See line 138)

Results:

  1. A sample description (e.g. mean age at survey, min/max age, exact time since diagnosis, stage at diagnosis, metastasis, recurrence, education, job situation, marriage status, living situation (alone/ with children…)) is missing. Were such data assessed?

Response: Thank you for your comments. (See lines 177-181)

  1. In l. 155 “183 mastectomized” relates to the whole sample of which however only 112 had mastectomy if I get it correctly.

Response: Thank you for your comments. An error was in he sample number. 183 corresponds to the total sample, where 112 is the number of mastectomised women. The data have been corrected thorughout the manuscript.

  1. In the tables, the abbreviations should be explained. The asterisks for the explanation of symptom and functioning scales might be misleading as asterisks are usually used to show statistically significant differences. Maybe superscript letters might be used instead.

Response: Thank you very much for your comments. We have made these changes in the manuscript.

  1. A simultaneous consideration of the variables in regression analyses, and maybe a more differentiated grouping of age, might lead to much more interesting and interpretable results. The age group > 50 years summarizes a wild mixture of people striving for return to work, retired people (who usually do better) and very old people (who usually do worse again due to their general condition).

Response: Thank you very much for your commment what you mention is of great interest. Perhaps the objective did not express correctly the aim of this article, it has been modified to express it correctly. I would like to point out that the article does not aim to look for a significant association, but to determine, in a descriptive way, whether some of the groupings of variables can influence the quality of life of these people. (See lines 101-104)

Discussion/ Conclusion:

  1. As further limitations, the missing randomization/ lack of a response rate, and lack of control of covariates (as probably not assessed), should be considered.

Response: Thank you for your comments. We have made these hanges in the manuscript. Although we considered not to add control for covariates given the descriptive aspecto of the article. (See line 343)

We hope we have now answered all your comments and we are looking forward to hearing from you again.

Jessica Fernández Solana, OT, PT

Reviewer 2 Report

Thank you for your interesting contribution. Here are some suggestions aiming at promoting your article, as follows:

- INTRODUCTION

There is a good overview of the incidence and frequency of breast cancer worldwide. However, it lacks an understanding of patients’ issues. For example, body image, self-esteem, and intimate relationship are only cited. Please, explore better these aspects, which are fundamental in the assessment of the Quality of Life (See: McGannon et al., 2012; McGinty et al., 2016; Sebri et al., 2022)

- It misses the contribution of social support, which need to be considered when dealing with Quality of Life. See: Sebri et al., 2021

References

-       McGannon, K. R., and Spence, J. C. (2012). Exploring news media representations of women's exercise and subjectivity through critical discourse analysis. Qual. Res. Sport Exerc. Health 4, 32–50. doi: 10.1080/2159676X.2011.653503

-       McGinty, H. L., Small, B. J., Laronga, C., and Jacobsen, P. B. (2016). Predictors and patterns of fear of cancer recurrence in breast cancer survivors. Health Psychol. 35, 1–9. doi: 10.1037/hea0000238

-       Sebri, V., Durosini, I., Mazzoni, D., & Pravettoni, G. (2022). The Body after Cancer: A Qualitative Study on Breast Cancer Survivors’ Body Representation. International Journal of Environmental Research and Public Health19(19), 12515.

-       Sebri, V., Mazzoni, D., Triberti, S., & Pravettoni, G. (2021). The impact of unsupportive social support on the injured self in breast cancer patients. Frontiers in Psychology12.

2. MATERIALS AND METHOD

- Why do you administer only two questionnaires? Why have you excluded other questionnaires that could be related to Quality of Life, even not directly? Please, explain your choice of questionnaire selection

- What about the medical drug? Are there some participants who are under pharmacological side effects? Did you explore this point?

3. RESULTS

- Regarding age, results were divided as before and after being 50 years old. Why this choice? I am not so sure that other differences could be possible. I argue that 30-40-50-60 could show strong emotional and cancer-related differences. Please, explain this point starting from the literature

4. DISCUSSION

- Make the discussion more detailed, please. For example, I agree with the idea that younger women could be more affected by emotional issues due to low support and their social role. What about the relationship with their body? What are their future expectations? Is impossibility of being pregnant? Please, deep in depth this point, starting from the current literature

- Please, add a section related to future research

Author Response

Mrs. Jessica Fernández Solana

Department of health sciences

University of Burgos, Paseo Comendadores s/n.

Burgos, 09001, Spain

Tel. (+34) 947499108

Email: jfsolana@ubu.es

10-11-2022

IJERPH.  Subject: Submissions Needing Revision

Dear editor.

Thank you very much for inviting us to submit our response to reviewers for our manuscript (ijerph-2014157) entitled: “Quality of life in breast cancer survivors in relation to age, type of surgery and length of time since first treatment”

We have checked our manuscript according to the Academic Editor, the reviewers’ comments and the Journal requirements. We have also responded to some comments from reviewers point by point).

We would be very grateful if you could consider our manuscript to be published in your journal.

Yours sincerely,

Jessica Fernández Solana, OT, PT

  1. Response to Reviewer 2:

First of all, we would like to express our sincere gratitude for all comments and suggestions received from the Reviewer 2. This information has certainly enriched the text for its best understanding, thank you very much indeed. We have clarified the reviewer2’s questions. We have introduced the required changes both in our answers to the specific comments and in the final manuscript V2.

Thank you for your interesting contribution. Here are some suggestions aiming at promoting your article, as follows:

- INTRODUCTION

There is a good overview of the incidence and frequency of breast cancer worldwide. However, it lacks an understanding of patients’ issues. For example, body image, self-esteem, and intimate relationship are only cited. Please, explore better these aspects, which are fundamental in the assessment of the Quality of Life (See: McGannon et al., 2012; McGinty et al., 2016; Sebri et al., 2022)

- It misses the contribution of social support, which need to be considered when dealing with Quality of Life. See: Sebri et al., 2021

References

 -       McGannon, K. R., and Spence, J. C. (2012). Exploring news media representations of women's exercise and subjectivity through critical discourse analysis. Qual. Res. Sport Exerc. Health 4, 32–50. doi: 10.1080/2159676X.2011.653503

-       McGinty, H. L., Small, B. J., Laronga, C., and Jacobsen, P. B. (2016). Predictors and patterns of fear of cancer recurrence in breast cancer survivors. Health Psychol. 35, 1–9. doi: 10.1037/hea0000238

-       Sebri, V., Durosini, I., Mazzoni, D., & Pravettoni, G. (2022). The Body after Cancer: A Qualitative Study on Breast Cancer Survivors’ Body Representation. International Journal of Environmental Research and Public Health19(19), 12515.

-       Sebri, V., Mazzoni, D., Triberti, S., & Pravettoni, G. (2021). The impact of unsupportive social support on the injured self in breast cancer patients. Frontiers in Psychology12.

Response: Thank you very much for your comments. These aspects you mention are undoubtedly very important, and therefore it is worth mentioning them in order to contextualise the problems surrounding the pathology of these patients in a broad way. Likewise, several variables are taken into account in the evaluation used in the study. However, this study wants to emphasise and give a descriptive picture of three very important variables in the context of breast cancer that can be influential in the quality of life of patients. (See lines 73-80)

  1. MATERIALS AND METHOD

- Why do you administer only two questionnaires? Why have you excluded other questionnaires that could be related to Quality of Life, even not directly? Please, explain your choice of questionnaire selection

Response: Thank you very much for pointing this out, we have added this information in the manuscript (See lines 131-133)

- What about the medical drug? Are there some participants who are under pharmacological side effects? Did you explore this point?

Response: Thank you very much for pointing this out, we have added this information in the manuscript (See line 344-346)

  1. RESULTS

- Regarding age, results were divided as before and after being 50 years old. Why this choice? I am not so sure that other differences could be possible. I argue that 30-40-50-60 could show strong emotional and cancer-related differences. Please, explain this point starting from the literature

Response: Thank you very much for your comments. This choice has been justified on the basis of the existing literature (See line 101-104)

  1. DISCUSSION

- Make the discussion more detailed, please. For example, I agree with the idea that younger women could be more affected by emotional issues due to low support and their social role. What about the relationship with their body? What are their future expectations? Is

impossibility of being pregnant? Please, deep in depth this point, starting from the current literature

Response: Thank you very much for your comments. We have added information regarding this in lines 287-290, 293-295 and 306-312.

- Please, add a section related to future research

Response: Thank you very much for your comments. We have added information regarding this in lines 349-356.

We hope we have now answered all your comments and we are looking forward to hearing from you again.

Jessica Fernández Solana, OT, PT

Round 2

Reviewer 1 Report

Dear authors, dear Editor,

thank you for the opportunity to review an updated version of the manuscript. The authors have put effort into the manuscript in order to improve the reading flow, the background and the description of the sample. However, my main concern, the calculation of a series of univariate tests without adjustment to potential covariates, has not been adressed. The authors have adjusted the goals of the study but did not change any results and state that they do not plan to do so. I think that the interpretation of such results is very difficult and does not add too much information to the existing literature. Operation method, age and time since diagnosis are likely to be correlated and we cannot entangle them in univariate analyses. The authors state repeatedly in the changed text that they had analyzed "the effect" or "influence" of the independent factors, but in a cross-sectional study this is not possible.

General comment: For future submissions/ reviews I would ask the authors to provide point-by-point responses to the review, not just per section, and including the lines of the changes as well as a quotation of the changed text. The updated manuscript should contain all changes, not only new text but also deleted text or changed tables. This makes it much easier for the reviewer and editor to follow the process.

Author Response

Mrs. Jessica Fernández Solana

Department of health sciences

University of Burgos, Paseo Comendadores s/n.

Burgos, 09001, Spain

Tel. (+34) 947499108

Email: jfsolana@ubu.es

30-11-2022

IJERPH.  Subject: Submissions Needing Revision

Dear editor.

Thank you very much for inviting us to submit our response to reviewers for our manuscript (ijerph-2014157) entitled: “Quality of life in breast cancer survivors in relation to age, type of surgery and length of time since first treatment”

We have checked our manuscript according to the Academic Editor, the reviewers’ comments and the Journal requirements. We have also responded to some comments from reviewers point by point).

We would be very grateful if you could consider our manuscript to be published in your journal.

Yours sincerely,

Jessica Fernández Solana, OT, PT

  1. Response to Reviewer 1:

First of all, we would like to express our sincere gratitude for all comments and suggestions received from the Reviewer 1. This information has certainly enriched the text for its best understanding, thank you very much indeed. We have clarified the reviewer1’s questions. We have introduced the required changes both in our answers to the specific comments and in the final manuscript V2.

Dear authors, dear Editor,

thank you for the opportunity to review an updated version of the manuscript. The authors have put effort into the manuscript in order to improve the reading flow, the background and the description of the sample. However, my main concern, the calculation of a series of univariate tests without adjustment to potential covariates, has not been adressed. The authors have adjusted the goals of the study but did not change any results and state that they do not plan to do so. I think that the interpretation of such results is very difficult and does not add too much information to the existing literature. Operation method, age and time since diagnosis are likely to be correlated and we cannot entangle them in univariate analyses. The authors state repeatedly in the changed text that they had analyzed "the effect" or "influence" of the independent factors, but in a cross-sectional study this is not possible.

Response: Thank you for your comments. We are sorry for having misstated the aim of our study, it has been modified to suit the statistical analyses performed and what was initially intended with this study (see lines 16-17, 97-99 and 261-263).

"The aim of this study was to describe the characteristics of the sample and to verify the relationship between QOL in female BC survivors and their age, type of surgery and time since first treatment".

The aim was to perform descriptive statistics, using univariate and bivariate analyses to describe, summarize and analyze the data relating to a characteristic of the individuals in a population And also,  individually to see if variables such as age, type of surgery or time since first treatment were related to the level of quality of life in these patients (taking into account both their symptoms and their functionality) (See lines 102 and 167-170).

General comment: For future submissions/ reviews I would ask the authors to provide point-by-point responses to the review, not just per section, and including the lines of the changes as well as a quotation of the changed text. The updated manuscript should contain all changes, not only new text but also deleted text or changed tables. This makes it much easier for the reviewer and editor to follow the process.

Response: Thank you very much for your comment. It will be taken into account for future submissions/reviews.

We hope we have now answered all your comments and we are looking forward to hearing from you again.

Jessica Fernández Solana, OT, PT

Reviewer 2 Report

Thank you for this work. Please, change the Tables' captions from Spanish to English. Then, in my opinion is accepted for publication

Author Response

Mrs. Jessica Fernández Solana
Department of health sciences
University of Burgos, Paseo Comendadores s/n.
Burgos, 09001, Spain
Tel. (+34) 947499108
Email: jfsolana@ubu.es
30-11-2022
IJERPH. Subject: Submissions Needing Revision
Dear editor.
Thank you very much for inviting us to submit our response to reviewers for our manuscript (ijerph-2014157) entitled: “Quality of life in breast cancer survivors in relation to age, type of surgery and length of time since first treatment”
We have checked our manuscript according to the Academic Editor, the reviewers’ comments and the Journal requirements. We have also responded to some comments from reviewers point by point).
We would be very grateful if you could consider our manuscript to be published in your journal.
Yours sincerely,
Jessica Fernández Solana, OT, PT
B. Response to Reviewer 2:
First of all, we would like to express our sincere gratitude for all comments and suggestions received from the Reviewer 2. This information has certainly enriched the text for its best understanding, thank you very much indeed. We have clarified the reviewer2’s questions. We have introduced the required changes both in our answers to the specific comments and in the final manuscript V2.
Thank you for this work. Please, change the Tables' captions from Spanish to English. Then, in my opinion is accepted for publication
Response: Thank you very much for your comments. We have made these changes in the manuscript.
We hope we have now answered all your comments and we are looking forward to hearing from you again.
Thank you very much,
Jessica Fernández Solana, OT, PT
